# Phosphodiesterase Inhibition to Sensitize Non-Small-Cell Lung Cancer to Pemetrexed: A Double-Edged Strategy

**DOI:** 10.3390/cancers16132475

**Published:** 2024-07-06

**Authors:** Anna V. Ivanina Foureau, David M. Foureau, Cody C. McHale, Fei Guo, Carol J. Farhangfar, Kathryn F. Mileham

**Affiliations:** 1Translational Research, Levine Cancer Institute, Atrium Health, Charlotte, NC 28204, USA; carol.farhangfar@atriumhealth.org; 2Immune Monitoring Core Laboratory, Levine Cancer Institute, Atrium Health, Charlotte, NC 28204, USA; david.foureau@atriumhealth.org (D.M.F.); fei.guo@atriumhealth.org (F.G.); 3Molecular Targeted Therapeutics Laboratory, Levine Cancer Institute, Atrium Health, Charlotte, NC 28204, USA; cody.mchale@atriumhealth.org; 4Thoracic Medical Oncology, Levine Cancer Institute, Atrium Health, Charlotte, NC 28204, USA

**Keywords:** non-small-cell lung cancer, phosphodiesterase inhibitors, pemetrexed, cyclic nucleotide signaling

## Abstract

**Simple Summary:**

We evaluated the repurposing of a class of non-cancer drugs, phosphodiesterase inhibitors, to increase the potency of an anti-folate chemotherapy drug against non-small-cell lung cancer. We showed that this approach represents a double-edged strategy. While the use of phosphodiesterase inhibitors potentiated the killing of lung cancer cells by combination low-dose anti-folate chemotherapy, in half of the tumors evaluated, the benefit was lost when using higher doses of chemotherapy.

**Abstract:**

Phosphosidesterases (PDEs) are key regulators of cyclic nucleotide signaling, controlling many hallmarks of cancer and playing a role in resistance to chemotherapy in non-small-cell lung cancer (NSCLC). We evaluated the anti-tumor activity of the anti-folate agent pemetrexed (PMX), alone or combined with biochemical inhibitors of PDE5, 8, 9, or 10, against squamous and non-squamous NCSLC cells. Genomic alterations to PDE genes (PDE^mut^) or PDE biochemical inhibition (PDEi) can sensitize NSCLC to PMX in vitro (observed in 50% NSCLC evaluated). The synergistic activity of PDEi with PMX required microdosing of the anti-folate drug. As single agents, none of the PDEis evaluated have anti-tumor activity. PDE biochemical inhibitors, targeting either cAMP or cGMP signaling (or both), resulted in significant cross-modulation of downstream pathways. The use of PDEi may present a new strategy to overcome PMX resistance of PDE^wt^ NSCLC tumors but comes with important caveats, including the use of subtherapeutic PMX doses.

## 1. Introduction

Metabolic reprogramming is a hallmark of tumorigenesis [1] that can be targeted for treating non-small-cell lung cancer (NSCLC). Pemetrexed (PMX), a second-generation anti-folate agent, was approved in combination with platinum-based chemotherapy for treating non-squamous NSCLC in 2008 [2,3]. The clinical utility of antimetabolites, such as PMX and the fluoropyrimidines, is partly attributed to their ability to abrogate the metabolic demand of cancer cells in nucleotide biosynthesis and DNA replication [4]. PMX is a folate analog that inhibits the one-carbon transfer reaction required for de novo nucleotide synthesis and thereby hinders tumor cell growth and promotes cell death by apoptosis [5,6].

Impaired cyclic nucleotide generation has been observed across cancer types. In lung cancer, for example, the altered expression and activity of cyclic nucleotide signaling mediators has been reported [7,8,9,10]. The growth and survival of cancer cells can be modulated by both cyclic adenosine monophosphate (cAMP)/ protein kinase A (PKA) and cyclic guanosine monophosphate (cGMP)/ protein kinase G (PKG) signaling [9]. Cyclic nucleotides cAMP and cGMP are generated from either ATP or GTP by the anabolic enzymes adenylate cyclase or guanylate cyclase, respectively. Once produced, cAMP and cGMP can be rapidly hydrolyzed by catabolic enzymes called phosphodiesterases (PDEs) that break the phosphodiester bond and inactivate cyclic nucleotides [11,12]. Eleven distinct PDE isozymes catalyze the breakdown of cAMP and/or cGMP [13,14]. Specifically, PDEs 5, 6, and 9 are selective for cGMP; PDEs 4, 7, and 8 are selective for cAMP; and PDEs 1, 2, 3, 10, and 11 are dual-substrate-degrading isozymes [15]. The inactivation of PDEs increases the levels of intracellular cyclic nucleotides which activate either PKA or PKG [15].

Through our published [16] and unpublished research, we have found that mutations in *PDEs* 8, 9, and 10 are associated with positive outcomes for patients with NSCLC. In addition, results of in vitro studies have shown that inhibition of PDEs blocks cell proliferation or induces apoptosis in NSCLC cells [17], glioblastoma stem-like cells [18], and other types of cancer cells [19,20,21]. In 2012, Nagai et al. demonstrated that in lung cancer cells, the toxicity of PMX is enhanced by increased nitric oxide levels via cGMP signaling [22]. DNA damage triggered by PMX was shown to activate protein kinase B (AKT), arresting cell growth in the S-phase and triggering apoptosis [4]. AKT is a proto-oncoprotein and the central downstream effector molecule of the PI3K pathway. In vitro data showed that AKT cooperated synergistically with both PKA and PKG to promote apoptosis [23].

Taken together, we sought to determine whether the biochemical inhibition of PDEs 8, 9, 10, and 5 could enhance PMX toxicity in lung cancer cells. Although PDE5 mutations are not associated with overall survival in patients with lung cancer, we decided to test the effect of PDE5 inhibition because the PDE5 inhibitor sildenafil is one of the most studied PDE inhibitors across multiple solid tumors [24], broadening the potential impact of this investigation. In this study, we evaluated the putative synergistic activity of four PDE inhibitors (PDEis 5, 8, 9, and 10) with PMX against both squamous and non-squamous NSCLC cells lines with intact purinergic metabolic pathway in vitro. We showed that PMX’s single-agent activity is associated with cAMP/cGMP catabolism. Genomic alterations in PDE genes or the biochemical inhibition of PDE function can sensitize NSCLC to PMX in vitro, independently of lung cancer histology (adenocarcinoma (AD) or squamous-cell carcinoma (SCC)). Further, we showed that highly selective PDE biochemical inhibitors, targeting cAMP/PKA and/or cGMP/PKG signaling, resulted in significant cross-modulation of downstream pathways. As single agents, PDEis have no cytotoxic or cytostatic effect on six selected PDE^wt^ NSCLC cell lines. In combination with PMX, we showed PDEi decrease lung cancer cell growth and trigger apoptosis in three out of six lung cancer cell lines tested. This effect was independent of NSCLC histology or baseline cAMP/PKA and cGMP/PKG signaling. Furthermore, we observed that the synergistic activity of PDEi/PMX was only present when in combination with microdosing of PMX, whereas PDEi in combination with higher dosing of PMX stimulated cell growth.

## 2. Material and Methods

### 2.1. Drugs and Reagents

Pemetrexed was purchased from Fisher scientific (Waltham, MA, USA). PDE inhibitors (PDEis) PF-04671536, PF-04447943, and PF-2545920, as well as sildenafil citrate and cisplatin, were obtained from Sigma Aldrich (St. Louis, MO, USA). PKA and PKG activators 8-Bromo-cAMP, sodium salt, and 8-Bromo-cGMP, sodium salt, were purchased from Tocris (Minneapolis, MN, USA). DMSO was used as a vehicle for all compounds unless otherwise noted. Trypsin–EDTA, EMEM, RPMI, and penicillin–streptomycin were purchased from ATCC (Manassas, VA, USA). Isozyme-specific antibodies (#GTX14625; GTX118886; GTX14621; GTX16418) for PDE isoforms (PDE9A; PDE10A; PDE8b; PDE8a) were purchased from GeneTex (Irvine, CA, USA). Other antibodies, including PDE5 (#2395) and phospho-specific antibodies for protein phosphorylation and their non-phosphorylated forms, were purchased from Cell Signaling technology (Danvers, MA, USA): these included VASP(9A2) (#3132); Phospho-VASP(Ser157) (#3111T); Phospho-VASP(Ser239) (# 3114); PKA Ca (# 4782); PKG-1(C8A4) (#3248S); Akt (#9272) and Phospho-Akt(Ser473)(D9E) (#4060) and β-actin (#4967). ADCY9(EPR16188) and GUCY1B3(EPR8822) were purchased from Abcam(Waltham, MA, USA), and secondary antibody from Jackson ImmunoResearch laboratories (West Grove, PA, USA).

### 2.2. Cells and Cell Culture

To choose cell lines without mutations in the cyclic nucleotide-related part of the purine metabolic pathway, we used the online database GEmiCCL (Gene Expression and Mutations in Cancer Cell Lines) (https://www.kobic.kr/GEMICCL/, accessed on 6 September 2020) (Appendix A). Three lung cancer AD and three lung cancer SCC PDE^WT^ cell lines were selected: cell lines NCI-H1781, Calu-3, NCI-H1373, NCI-1703, and NCI-H226 were purchased from ATCC (Manassas, VA, USA), and Ludlu-1 was purchased from Sigma Aldrich (St. Louis, MO, USA). Cells were cultured under standard cell culture conditions in RPMI 1640 or EMEM (for Calu3) mediums containing 10% fetal bovine serum and supplemented with 2 mM L-glutamine, 10 IU per mL penicillin, and 10 µg per mL streptomycin at 37 °C in a humidified atmosphere with 5% CO_2_. BEAS-2B cells, a normal human bronchial epithelium cell line, were provided by Dr. Donald Durden’s Molecular Targeted Therapeutics Laboratory at the Levine Cancer Institute and cultured according to the manufacturer’s instructions in complete Airway Epithelial Cell Basal Medium (ATCC) on fibronectin/bovine collagen I/bovine serum albumin-coated plates. All cell lines were expanded upon delivery and numerous vials of low-passage cells were preserved in liquid nitrogen. Cells were passaged in culture for no longer than two months. Cell-line characterization was conducted by ATCC and Sigma Aldrich. No additional authentication of cell lines was conducted except for experimental reasons (e.g., confirmation of protein expression level and sensitivity to biochemical inhibitors).

### 2.3. Western Blotting

For protein phosphorylation studies, whole cell extract was obtained by using ice-cold cell lysis buffer (BioVision, Milpitas, CA, USA) supplemented with a protease inhibitor cocktail (#K272, BioVision) and 10 mmol/L NaF (Sigma Aldrich). Cells were centrifuged for 15 min at 10,000× *g* and 4 °C. Protein concentration was determined using the Bio-Rad Protein Assay kit (#5000002, Bio-Rad Laboratories, Hercules, CA, USA) and 30 µg per µL of cell lysate. Cell lysates were separated by 4–12.5% gradient pre-casted SDS-PAGE gels (Bio-Rad Laboratories) followed by electrophoresis transfer to a nitrocellulose membrane. Equal loading and transfer were verified with Ponceau staining (Fisher Scientific, Waltham, MA, USA). The membranes were blocked in 5% bovine serum albumin in Tris-buffered saline and probed with primary monoclonal antibodies against PDE5a, 8a and 8b, 9a, and 10a; PKA; PKG; ADCY; GUCY; VASP; pVASP157; pVASP239; AKT; pAKT; and β-actin. Western blotting procedures were performed using Chemiluminescence Reagent (Fisher Scientific) and visualized using Imaging System VisionWorks (Analytic Jena US, Upland, CA, USA). All antibodies produced bands of the expected molecular size. Densiometric analysis of the signal was performed using VisionWorks software version 8.21 (Analytic Jena US). For semi-quantitative analysis, band intensity was normalized to that of β-actin for each treatment.

### 2.4. Growth Assays

To measure cell growth, triplicate wells of 96-well plates were seeded at a concentration of 5000 cells per well and allowed to recover for 4 h (h) at 5% CO_2_ at 37 °C. Then, the cells were treated with the indicated doses of drugs or DMSO (final concentration at 0.5%) in 96-well tissues culture plates and incubated for 72 h in 5% CO_2_ at 37 °C. Cell growth was determined by WST-8 assay using the Cell Counting Kit-8 (#Ab228554, Abcam)according to the manufacturer’s instructions. Briefly, after 48 h of treatment, 10 µL of 2-(2-methoxy-4-nitrophenyl)-3-(4-nitrophenyl)-5-(2,4-disulfophenyl)-2H-tetrazolium monosodium salt (WST-8) reagent was added to each well and incubated at 37 °C for 24 h. The absorbance at 460 nm was measured using BMG Labtech microplate reader (BMG Labtech, Ortenberg, Germany). Cell numbers after the treatment were calculated based on cell dilution standard, which were run with every experiment.

### 2.5. Cell Viability and Apoptosis Assays

To measure cell viability, duplicate wells of 12-well plates were seeded at a concentration of 5000 cells per well and allowed to recover for 4 h at 5% CO2 at 37 °C. The cells were then treated with the indicated doses of drugs or DMSO for 72 h. Cell viability was measured on a Muse cell analyzer (CyTek Biosciences, Fremont, CA, USA) using Propidium Iodide assay (#SKU MCH100102, Muse Count & Viability kit, CyTek Bioscience) according to manufacturer’s instructions. Cellular apoptosis was measured on a BD lsrFortessa Flow cytometer (BD Biosciences, San Jose, CA, USA) by performing 7-7-Aminoactinomycin D (7-AAD)/Annexin-V staining as previously described [25].

### 2.6. Cell Proliferation Assay

To measure cellular proliferation, cells were treated with the indicated doses of drugs or DMSO for 24 h. The cell cycle (G1-S-G2 phase distribution) was evaluated using Vybrant™ DyeCycle Violet (#C10094, Invitrogen, Eugene, OR, USA) as previously described [26].

### 2.7. cAMP and cGMP Measurements

Cells were treated with either PDEi, PMX, or a combination of both for 72 h. Cells were incubated with 0.1 M HCl for 20 min and centrifuged at 1000× *g* for 10 min and stored at −80 °C until further analysis. The concentration of cAMP/cGMP was determined using a cAMP/cGMP kit according to manufacturer’s recommendations (#581001/#581021, Cayman Biochemical, Ann Arbor, MI, USA). In brief, samples were acetylated using 4 M KOH and acetic anhydride and incubated with either cAMP or cGMP acetylcholinesterase conjugate tracer and with cAMP/cGMP ELISA antiserum for 18 h at +4 °C. Signal was developed using Ellman’s reagent with 90 min incubation and measured at 405 nm using BMG Labtech microplate reader. The concentration of cAMP/cGMP was calculated based on the cAMP/cGMP standard and expressed as cAMP/cGMP pmol per mL.

### 2.8. PKA and PKG Activity Assays

Cells were treated with PDEi for 72 h. Cells were trypsinized and briefly centrifuged; the supernatant was snap-frozen in liquid nitrogen and stored at −80 °C until further analysis. PKA and PKG activities were determined, respectively, using a PKA kinase activity assay kit (#ab139435, Abcam) and a CycLex Cyclic GMP-dependent protein kinase assay kit (#CY-1161, MBL International Corporation, Woburn, MA, USA). The frozen cells were lysed in a buffer containing 0.5 µL per mL of protease inhibitor cocktail (#K272, BioVision), 1 mM PMSF, and 10 mmol per L NaF (Sigma Aldrich) and centrifuged at 10,000× *g* for 10 min at 4 °C. Using the supernatant, assays were performed following the respective manufacturers’ protocols. Protein concentrations were measured using Bio-Rad Protein Assay kit. For PKG activity, cGK positive control (#CY-E1161-2, full length; MBL International Corporation) was used as a standard. Results were expressed as U/µg protein.

### 2.9. Apo-Tox-Glo Triplex Assay and Drug Interaction Analysis

Following manufacturer instructions, the ApoTox-Glo Triplex assay (#G6321, Promega, Madison, WI, USA) was used to determine the effect of either PDEi, PMX, or of the combination of both on cell viability, cytotoxicity, and apoptosis. Fluorescence and luminescence were measured with a BMG Labtech microplate reader (BMG Labtech).

Multi-drug combination anti-NSCLC activity, calculated using normalized cell counts (%DMSO control) adjusted with apoptotic rates, was used as input for multi-drug interaction analysis on SynergyFinder version 3.0 (https://synergyfinder.fimm.fi/) [27]. A combined (all PDEi) Bliss synergy score was calculated for each NSCLC cell line sensitive to a PMX-PDEi combination (Ludlu1, H1703, H1781). Bliss scores were also calculated individually for PDEi in combination with low (0.025 µM) or high (250 µM) PMX concentrations.

### 2.10. Statistical Analysis

Statistical analysis was performed using JMP 17 software (SAS Institute, Cary, NC, USA) and graphs were made using GraphPad Prism v10.0 (GraphPad Software Inc., La Jolla, CA, USA). Descriptive statistics, including means, standard error (SEM), counts, and percentages were calculated. All N, *p*-value and F, df values are mentioned in the respective figures, tables, or associated legends. The effects of PDEi, PMX, and their combination on cellular parameters were assessed using generalized linear model ANOVA. Both factors (PDEi and/or PMX) were treated as fixed. Post hoc tests (Fisher’s least square difference) were used to test the differences between the group means. Unless otherwise indicated, data are represented as means ± SEM. The differences, unless otherwise indicated, were considered significant if the probability of Type I error was less than 0.05. Spearman’s correlations were used to test for linear relationships between variables and increased concentrations of PDEi or PMX. In this study, we used a hierarchical cluster analysis (Ward’s methods) using determined measurements (cAMP, cGMP, pVASP157, pVASP239, and pAKT) recorded for each cell line. We showed a dendrogram to estimate the number of likely clusters. Clusters were visualized using a heat map.

## 3. Results

### 3.1. Baseline cAMP and cGMP Signaling in NSCLC Cell Lines

In each of the six NSCLC cell lines, we measured the expression levels of catabolic enzymes (PDE isoforms 8a, 8b, 5a, 9a, and 10a) and anabolic enzymes (adenylate cyclase, ADCY, and guanylate cyclase, GUCY) involved in cyclic nucleotide metabolism (Figure 1A,B). PDE8b expression was highest in the SCC Ludlu1 cell line and lowest in the AD H1781 cell lines, whereas the expression of the PDE8a, PDE9a, and PDE10a isoenzymes was ubiquitous and abundant in all cell lines. PDE5a expression was heterogenous, detected at higher levels in H1373, Calu3, and H226 cells and lower levels in H1781 and Ludlu1. H1703 cells displayed only trace PDE5a expression. The baseline expression of ADCY and GUCY was opposite among the studied cell lines. Four cell lines (H1373, H1703, H226, and Ludlu1) had high levels of expression of ADCY and simultaneously low GUCY (Figure 1A,B). Baseline cAMP levels differed between cell lines, with the highest cAMP concentration observed in H1373 (0.789 pmol per µg protein) and the lowest cAMP concentration (0.168 pmol per µg protein) in Ludlu1 (F_5,15_ = 20.524, *p* < 0.0001). The concentration of cGMP was similar among cell lines (Figure 1C; F_5,20_ = 1.133, *p* = 0.385). We evaluated baseline cyclic nucleotide signaling in NSCLC by measuring PKA and PKG activity. H1373 had the highest level of PKA and PKG activity, whereas the PKA activity of the Calu3 and H1703 cell lines was 5.9–9.9 times lower than that of the other four cell lines. Ludlu1 cells had significantly lower PKG activity than other cell lines (0.1 vs. 0.18–0.23 U per µg protein, respectively) (Figure 1C; F_5,11_ = 89.350, *p* < 0.0001; F_5,11_ = 28.934, *p* = 0.0004 for PKA and PKA, respectively). Baseline level markers of PKA and PKG activity (pVASP157 and pVASP239, respectively) differed between cell lines but not between histological categories (AD vs. SCC, Figure 1D). Taken together, we found no differences between the expression, cellular concentration, or downstream signaling of cAMP/cGMP metabolic enzymes on the basis of lung cancer histology.

### 3.2. PDEi Modulates Cell Growth and Cyclic Nucleotide Signaling in NSCLC

All PDE inhibitors investigated in the study (PDE8i, PDE5i, PDE9i, and PDE10i) promoted cell growth in a dose-dependent manner across all evaluated NSCLC cell lines, except Ludlu1 (Appendix A, Appendix A). PDE8i and PDE10i were the most potent agents in promoting cell growth (PDE8i EC_50_ 3.7 ± 0.65 nM; PDE10i EC_50_ 1.7 ± 0.55 nM), while sildenafil (PDE5i) required micromolar dosing to affect cells phenotypically (PDE5i: 2698 ± 648.38 nM) (Appendix A). When tested against normal human bronchial epithelium cells BEAS-2B, the biochemical inhibition of PDE5, 8, or 9 did not significantly alter cell growth (Appendix A). Only PDE10i lowered BEAS-2B cell counts in a dose-dependent fashion after 72 h treatment (at 6–10 nM, F_1,41_ = 9.6515, *p* < 0.0001). This PDE10i effect was not associated with any cytotoxic activity toward BEAS-2B cells as shown by the absence of cell death or apoptosis (Appendix A). Nor was there cytostatic activity, as shown by the absence of cell cycle arrest (Appendix A). Overall, single-agent PDEi showed no anti-tumor activity against NSCLC cell lines or toxicity toward normal lung epithelial cells.

We evaluated PDEi target activity by measuring intracellular levels of cAMP and cGMP after treatment with PDEi (Figure 2A). The cellular concentration of cAMP increased in response to PDE8i in five of the six cell lines, while cGMP increased in response to PDE5i in four cell lines and in response to PDE9i and PDE10i in five cell lines. Together with PDEi target activity, we also evaluated PDEi off-target activity and found that, in most instances, PDEi simultaneously modulated both cAMP and cGMP cellular concentration in NSCLC (Figure 2A,B). Lastly, we measured the indirect effect of PDEi on cAMP/cGMP anabolic enzyme expression (ADCY or GUCY, Figure 2B and Appendix A). PDE8i inhibited ADCY expression in all six cell lines, while off-target activation GUCY was observed in H226 cells only. PDE5i increased GUCY levels in one cell lines (Calu3); PDE9i and PDE10i increased GUCY levels in two cell lines (H226 and H1781 and Calu3 and H1373, respectively). Off-target effects of PDEi were also observed on ADCY and GUCY levels. PDE5i increased ADCY levels in three cell lines, PDE9i increased ADCY levels in two cell lines, and PDE10i increased ADCY levels in four cell lines. None of the inhibitors had off-target effects on the expression of ADCY or GUCY in H1373 and H226 cells. Interestingly, H1781 exposure to PDE5i or PDE10i was associated with the simultaneous inhibition of protein expression of GUCY and stimulation of ADCY (Figure 2B and Appendix A).

We then evaluated PDEi modulation of cAMP/PKA and cGMP/PKG signaling in NSCLC by measuring the phosphorylation of the downstream target vasodilator-stimulated protein (VASP, Figure 2C and Appendix A). The phosphorylated forms of VASP pVASP157 and pVASP239 have established cAMP-activated PKA or cGMP-activated PKG phosphorylation sites, respectively [28,29]. Relative to expression in untreated cells, PDE8i increased pVASP157 expression in three cell lines and inhibited pVASP157 expression in the other three. Simultaneously, the cross-inhibition of pVASP239 (PKG activity) was observed in five cell lines, but in H226 cells, pVASP239 was increased. PDE5i, PDE9i, and PDE10i increased pVASP239 in Calu3 AD cells and two SCC cell lines, H226 and Ludlu1. By contrast, in other cell lines, we observed decreased protein expression of PKA and PKG targets (Figure 2C, Appendix A). Since we observed inconsistency in PKA/PKG downstream target phosphorylation triggered by PDEi across NSCLC cell lines, we decided to investigate the cellular response to direct PKA/PKG stimulation in each cell line (Figure 2D and Appendix A). The PKA activator 8-bromo-cAMP at 0.05 mM promoted pVASP157 phosphorylation in three cell lines (H1781, H1703, and Ludlu1: 1.52 ± 0.25-fold change from baseline) but decreased pVASP157 phosphorylation in the other two cell lines (H1373 and H226: 0.81 ± 0.04-fold change from baseline). We also observed cross-PKG modulation by 8-bromo-cAMP, with a pVASP239 increase also observed in three cell lines (H1781, H1703, and Ludlu1: 2.06 ± 0.46-fold change from baseline) and a decrease in the remaining three cell lines (H226, H1373, and Calu3: 0.92 ± 0.06 -fold change from baseline). No further changes in PKA or PKG activity were observed at higher 8-bromo-cAMP concentrations. At 0.05 mM, the PKG activator 8-bromo-cGMP triggered a more consistent pVAP239 increase across all cell lines (1.57 ± 0.18-fold increase) except H226 and demonstrated a significant cross-activation of pVASP157 (1.24 ± 0.15-fold increase). PKA/PKG activation was abrogated at higher 8-bromo-cGMP concentrations (0.1–1 mM). Recognizing that the concentration range of 8-bromo-cAMP and 8-bromo-cGMP used in the activation experiments far exceed physiological concentrations of intracellular cAMP (0.53 ± 0.03 pmol per µg protein) and cGMP (0.21 ± 0.01 pmol per µg protein) measured in NSCLC cell lines treated with PDE inhibitors, we next evaluated whether increased intracellular cyclic purine nucleotides can modulate the phosphorylation of both PKA and PKG targets under physiological conditions (Appendix A). Linear regression analyses of intracellular cAMP content measured combining all PDEi treatments showed limited modulation of pVASP157. However, a positive correlation between cAMP content and pVAP239 was established for three cell lines (H1373, Calu3, and Ludlu1, *p* < 0.2). Similarly, a positive correlation between cGMP cellular content and both pVASP157 and pVASP239 phosphorylation was established in Ludlu1 cells (*p* < 0.2). This effect was not observed in all cell lines, since H1781 and H226 cGMP content positively correlated with PKG pVASP239 phosphorylation only (*p* < 0.2), whereas H1373 cGMP content negatively correlated with PKA target pVAP157 phosphorylation (*p* < 0.001) (Appendix A).

An unsupervised hierarchical cluster analysis was applied to all samples (six NSCLC cell lines, exposed to four PDEis each) evaluable for cAMP and cGMP concentrations and PKA/PKG indirect activity (pVASP157 and pVASP239 protein expression, respectively) (Figure 2E). Each cell line, rather than individual PDE inhibitors or histological categories, was clustered together. Four cell lines (Calu3, H1781, H1373 and H226) were clustered together. The common features of these cell lines in the presence of PDEi were a low cGMP level and no activation of PKG (low protein expression of pVASP239). H1703 and Ludlu1 formed two separate clusters. In the presence of all four PDEis, the H1703 cluster was characterized by a low level of cGMP and low activation of PKA (phosphorylation of VASP157), while Ludlu1 was characterized by a low concentration of cAMP and increased concentration of cGMP.

### 3.3. PMX Anti-NSCLC Activity Associates with Modulation of cAMP/cGMP Cellular Content and Downstream Signaling

Single-agent PMX displayed a pleiotropic growth-stimulating activity toward all six cell lines (Figure 3A, Appendix A). To confirm the lack of single-agent PMX cytotoxic activity toward the PDE^wt^ NSCLC cell lines evaluated in our study, we performed additional viability and apoptosis measurements of cells exposed to PMX alone or in combination with cisplatin (Figure 3B,C). Single-agent PMX had no (or only marginal, <2%) cytotoxic activity toward all tested AD and SCC cells (Figure 3B). Importantly, when combined with cisplatin, PMX significantly enhanced platinum cytotoxicity in a dose-dependent fashion, confirming PMX’s activity in combination but not as a single agent in the models evaluated in our study. Confirming those findings, PMX was required to be combined with cisplatin to induce an apoptotic state (7-AAD^+^, Annexin-V^+^) (Figure 3B). The only notable exception was H226 cells, which achieved 8.8% apoptotic rate with high-dose PMX alone (11.98 ± 0.61 vs. 20.83 ± 1.93% for control and 250µM PMX, respectively). Single-agent high-dose PMX (250µM) also led to a dramatic increase in H1781, Calu3, and Ludlu1 cells, which entered a pre-apoptotic rate (7-AAD^−^ Annexin-V^+^), a reversible physiological state akin to autophagy–senescence (Figure 3C). This effect was most pronounced in H1781 cells, where almost 26.14% cells entered this reversible pre-apoptotic state after high-dose PMX exposure (9.76 ± 0.22 vs. 35.9 ± 1.48% for control and 250 µM PMX, respectively).

Beyond its antimetabolite function, PMX can activate the AKT pathway via phosphorylation of AKT [30,31,32]. Therefore, we evaluated AKT activation in response to PMX treatment in all six cell lines. Although PMX triggered AKT activation (pAKT) in a dose-dependent fashion among two of three AD cell lines, PMX had the opposite effect among two of three SCC cell lines, with only Ludlu1 displaying a transient pAKT increase (Figure 4A and Appendix A). We evaluated whether PDEi similarly modulated pAKT in a dose-dependent fashion. PDE8 inhibition indeed promoted pAKT in two of three AD cell lines and decreased pAKT in two of three SCC cell lines (Figure 4A and Appendix A). Of the six cell lines, only Ludlu1 displayed transient pAKT activation in response to all four PDE inhibitors. Therefore, we tested whether direct PKA or PKG activation modulated pAKT. Whereas 8-bromo-cAMP promoted activation in the two AD cell lines, H1373 and H1781, 8-bromo-cGMP decreased the phosphorylation of AKT (Figure 4A and Appendix A). Similarly, whereas direct PKA activation diminished pAKT levels in the remaining four NSCLC cell lines, direct PKG activation had the opposite effect. Overall, we observed that independent of NSCLC etiology, PKA and PKG activation does indeed modulate pAKT, but with opposing effects.

Next, we evaluated the effect of PMX on cAMP/PKA and cGMP/PKG signaling. The antimetabolite tended to decrease cAMP cellular content, which was more pronounced with the two AD cell lines, H1373 and H1781 (Figure 4B). In all but one NSCLC cell lines, PMX led to a sharp decrease in cellular cGMP at concentrations as low as 0.05–0.11 pmol per µg protein (Figure 4B,C); only Ludlu1 retained high cGMP levels even with high-dose PMX (0.23 vs. 0.26–0.29 pmol per µg protein at baseline vs. PMX-treated, respectively). The effect of cAMP/cGMP cellular content on downstream PKA/PKG signaling was inconsistent across cell lines and was not correlated with NSCLC histology. In two AD cell lines (H1781 and Calu3), PMX increased levels of the PKA target pVASP157 while inhibiting the PKG target pVASP239. We observed the opposite trend (i.e., pVASP157 decrease matched with a pVASP239 increase) in two other cell lines (H1373, H1703) (Figure 4C and Appendix A).

We conducted an unsupervised hierarchical cluster analysis on all samples (six cell lines treated with low or high concentrations of PMX, 0.025 and 250µM PMX, respectively) and evaluated them for cAMP and cGMP cellular content, PKA/PKG activity (pVASP157 and pVASP239 protein expression), and AKT activation (pAKT) (Figure 4D and Appendix A). Four cell lines (one AD cell line, H1373, and three SCC cell lines, H1703, H226, and Ludlu1) clustered together, sharing a low overall pAKT activation in response to PMX (Figure 4D). H1373 and H1703 also displayed high PKG activation (pVASP239). Ludlu1 showed no PKG increase despite cGMP increasing in response to PMX. Similarly, H1703 showed no PKA increase despite cAMP increasing in response to treatment. Two AD cell lines (H1781 and Calu3) formed a separate cluster characterized by high pAKT activation in response to treatment. Calu3 also showed strong PKA (pVASP157) upregulation.

Given PMX’s modulation of cAMP/cGMP cellular content and downstream signaling, as well as the lack of single-agent activity observed in our study, we further investigated whether cAMP/cGMP metabolism was associated with PMX’s anti-NSCLC activity. Earlier work performed by Dr. Misty Shields in Dr. John Minna’s laboratory at University of Texas Southwestern Medical Center described heterogeneous responses to PMX in a NSCLC cell line screen assay. Their work confirmed the PMX resistance of H226 and H1703 observed in our study and described a similar phenotype for three additional cell lines (H1373, H1975, and H2073). Together with our dataset, five AD cell lines and three SCC cell lines have demonstrated a PMX-resistant phenotype (Figure 4E). Dr. Shields also characterized eleven AD cell lines sensitive to PMX. We next evaluated genomic information available for those nineteen NSCLC cell lines. All PMX-sensitive NSCLC cell lines had at least one mutation in genes controlling cAMP/cGMP metabolism, with 64% of them containing mutations of cAMP catabolic genes (*PDE4B-C*, *PDE8B*), cGMP catabolic genes (*PDE5A*, *PDE6A-B*, *PDE6D*), or dual cAMP/cGMP catabolic genes (*PDE1A-C*, *PDE3A-B, PDE10A*) (Figure 4E). Contingency analyses showed that genomic alterations in cAMP/cGMP catabolic enzymes were strongly associated with PMX sensitivity (*p* = 0.0128) (Figure 4F). We also evaluated EGFR, ALK, KRAS, and p53 mutation status in those cell lines and saw no correlation with PMX sensitivity (Appendix A). Only NSCLC histology was associated with response, with SCC cells marginally more likely to be PMX-sensitive compared with AD cells (*p* = 0.0578); however, SCC cells were also significantly more likely to contain mutated cAMP/cGMP catabolic genes (*p* = 0.0361).

### 3.4. Biochemical PDE Inhibition Sensitizes NSCLC to Low-Dose PMX Killing

Although single-agent PDEi and PMX promoted cell growth, combination PDEi and low-dose PMX (0.025 µM) decreased cellular growth in three of the cell lines tested, with no significant changes in cell growth in the remaining cell lines (Figure 5A, Appendix A). Cell growth inhibition in H1781, H1703, and Ludlu1 was PDEi-dose-dependent regardless of the PDE isoenzyme targeted (Appendix A). We further investigated mechanisms of cellular loss caused by combination PDEi with low-dose PMX and showed apoptosis was triggered in a PDEi-dose-dependent fashion when combined with 0.025 µM PMX in both the H1703 and Ludlu1 SCC cell lines (Figure 5A). All PDEis achieved 151.06 ± 5.17% H1703 and 166.67 ± 4.96% Ludlu1 cell killing at the highest dose tested (relative to untreated control). In the H1781 AD cell line, the combination PDEi with low-dose PMX did not promote cell death (apoptosis or necrosis), suggesting cytostatic activity.

When combined with high-dose PMX (250 µM), the cytotoxic effect of PDEi-PMX combination was largely diminished in the two SCC cell lines (Ludlu1, H1703) (Figure 5A). Multi-drug interaction analyses confirmed that the strong synergistic activity between PDEi and low-dose PMX, contributing to the killing of the two SCC cell lines, was significantly diminished with high-dose PMX (Figure 5B). This effect was mainly observed when high-dose PMX was combined with PDE5-9i and 8i (Figure 5B). The cytostatic activity of PDEi plus 0.025µM PMX combination toward AD cells H1781 reverted to a strong cell-growth-stimulating effect when 250µM PMX was employed. Multi-drug interaction analyses showed weak synergy between PDEi and low-dose PMX but strong antagonistic activity when high-dose PMX was combined with either PDE8i, PDE5-9i or PDE10i (Figure 5A,B)

We conducted an unsupervised hierarchical cluster analysis that included all samples (six cell lines treated with combination PDEi-PMX) and evaluated for cAMP and cGMP cellular content, PKA and PKG activity (via pVASP157 and pVASP239 protein expression), and AKT activation (pAKT) (Figure 5C and Appendix A). Similar to what we observed with single-agent PDEi and PMX, each cell line clustered separately across all treatment groups (Figure 5C). Calu3, H1703, and H1781 shared high AKT activation independent of PDEi and PMX concentrations. PKA activation (pVASP157) in response to combination treatment was restricted to H1781 and H226 cells, whereas PKG activation (pVASP239) was restricted to Calu3 and H226. Although modulation of cAMP or cGMP cellular levels did not correlate with the respective PKA and PKG activation, we found a strong association between cAMP and PKG signaling in all but one cell line (H1703). We did not find consistent patterns between altered cAMP/PKA, cGMP/PKG, and pAKT activation by NSCLC etiology or response to combination PDEi-PMX.

## 4. Discussion

PDE inhibitors are increasingly being studied for the prevention and treatment of multiple types of cancer. For instance, in 2022, ClinicalTrials.gov recorded 28 clinical studies involving the use of PDE inhibitors to target solid malignancies [34]. The present study provides new insights on PDE inhibition in NSCLC and the potential synergy with the antimetabolite PMX. Our main findings demonstrate that despite a high affinity and nanomolar potency for their targets, biochemical inhibitors targeting either cAMP/PKA (PDE8i) or cGMP/PKG (PDE5i, 9i, 10i) cross-modulate each other. We identified three cross-reactivity levels that prevent the decoupling of cyclic purinergic nucleotide signaling and inhibition of PDE catabolic activity, including (1) feedback modulation of cAMP/cGMP anabolic enzymes, (2) PKA/PKG signaling interactions, and (3) cross-activation of other protein kinases, such as AKT (i.e., protein kinase B). Although we successfully sensitized classically resistant squamous NSCLC cells to PMX by combining PMX with PDEi, this effect was restricted to low-dose PMX (sub-therapeutic) and only observed in a subset of the evaluated cell lines. Dual PDEi-PMX treatment can be a double-edged strategy, as we observed that higher doses of PMX can turn the drug combination into a potent cell-growth-stimulating/anti-apoptotic regimen. The two main limitations of our study are (1) the use of a heterogenous cell line screen approach rather than the creation of isogenic cell lines to mechanistically evaluate PDEi-PMX interactions and (2) a lack of in vivo validation. The effects of PDEi on cyclic nucleotide signaling varied between NSCLC cells evaluated, suggesting biochemical inhibition of PDE does not have a universal impact on NSCLC.

Phosphodiesterase inhibition and modulation of downstream cAMP/PKA or cAMP/PKG signaling have been proposed for the treatment of NSCLC. For instance, PDE5i and 10i have shown promising anti-tumor activity as single agents in NSCLC [10,35]. Earlier work by our group showed that non-synonymous mutations in *PDE8*, *PDE9*, and *PDE10* are associated with longer overall survival of patients with squamous or non-squamous NCSLC, respectively [16]. Although we have confirmed ubiquitous expression of PDE8-10 in both squamous and non-squamous cell lines, PDE5 expression was far more heterogeneous ranging from a faint signal to very high expression. This raises a question about whether PDE5 is a suitable target in NSCLC and may explain why PDE5i required micromolar concentrations to effect NSCLC cells rather than nanomolar concentrations used with PDE8i-10i. Regardless of the choice of PDE target or PDEi affinity for their target, our findings indicate that modulation of cellular cAMP or cGMP are difficult to decouple. From an enzymatic activity standpoint, our data go hand-in-hand with those of a previous study on cardiac myocytes, where cGMP regulated cAMP-mediated signaling by modulating cAMP-hydrolyzing PDEs. Small elevations in cGMP levels were associated with the stimulation of cAMP-dependent PDEs, while higher cGMP levels inhibited cAMP-dependent PDEs activity [14]. We also showed that inhibiting cyclic purine nucleotide catabolism triggers the modulation of anabolic enzymes (adenylate and guanylate cyclases), further highlighting the difficulty of decoupling cAMP and cGMP metabolism.

Further compounding the difficulty of achieving specific cAMP/PKA or cGMP/PKG modulation using biochemical PDEi in NSCLC, we observed PKA/PKG cross-modulation even with their direct activators, 8-bromo-cAMP and 8-bromo-cGMP. Although PKA is the primary protein kinase activated by cAMP, several studies suggest that some of the biological effects of cAMP may be mediated by the activation of PKG [36,37,38,39]. A previous study also showed direct activation of PKA via the phosphorylation of PKA catalytic subunit in the absence of cAMP [40]. Cross-modulation of protein kinases is not restricted to PKA and PKG in NSCLC and extends to AKT. Similar to DNA-damaging agents [30,41], PDEis have been previously observed to activate AKT, for example, PDE4Bi in human diffuse large B cell lymphoma [42] and tadanafil (PDE5i) in murine ischemic injury [43]. Mechanistically, crosstalk between cAMP signaling and AKT has been previously proposed [42,44]. We also showed that PMX inconsistently affects PKA activation across NSCLC cell lines (gain of pAKT in 3/6 cell lines (AD H1373, Calu3, and SCC Ludlu1) and loss of pAKT in two SCC cell lines (H1703 and H226)). Such inconsistency reflects earlier findings by Bischof et al. and Chen et al., who both showed that PMX promotes AKT phosphorylation [30,45], whereas Tekle et al. observed decreased phospho-AKT levels in the A549 cell line [46]. Whether PMX modulates AKT activation directly or does so indirectly though cAMP/cGMP cellular content modulation remains to be determined. Indeed, to the best of our knowledge, our study is the first to show regulation of cyclic nucleotide cellular content by PMX.

A key observation in our study is that neither single-agent PDEi nor PMX had cytotoxic or cytostatic effect against the six NSCLC cell lines we evaluated. In all cells except Ludlu1, all PDE inhibitors (PDEs 8i, 5i, 9i, and 10i) stimulated cellular growth, with no discernable differences in the degree of response between individual inhibitors in each cell line. Those findings seem to contradict earlier work which showed that elevated cAMP or cGMP levels were associated with growth arrest or cell death in HT29 colon cancer cells [47], leukemia HL-60 cells [48], hepatoma Hep2G2 cells [49], and lung cancer cell lines A549, H23, and H522 [10]. Our findings align more closely with work by Naderi et al. on lymphocytes, which showed increased levels of cellular cyclic nucleotides associated with increased cellular proliferation [50]. Contradictory results in the literature may be partly attributed to differences in in vitro models employed across studies. For instance, lung cancer cell line H522 has three non-synonymous mutations affecting cyclic purine nucleotide metabolism: one in guanylate cyclase isoforms and two in PDEs. Similarly, A549 has one non-synonymous mutation in the *PDE4D* gene (Appendix A). Although rare, we have shown that PDE mutations are clinically impactful in non-squamous NSCLC [16]. Due to the high level of cAMP/cGMP metabolism and signaling cross-modulation uncovered in our study, selecting PDE-mutated cell lines may also introduce biases for evaluating biochemical inhibition. Such putative biases may extend to antimetabolite activity, since we also show that single-agent PMX has strong growth-stimulating activity in the NSCLC cell lines we evaluated. Our study strongly suggests that cAMP/cGMP catabolism (and associated downstream signaling) may in fact be a key component of NSCLC cells’ response to the antimetabolite, shedding a new light on conflicting single-agent PMX activity in vitro results described in the literature [33,51,52,53,54]. Rather than a single gene association, consistent with the high level of cAMP/cGMP cross-modulation observed, we found that genomic alterations (i.e., non-synonymous mutation) of a wide range of PDE genes, including cAMP catabolic genes (*PDE4B-C*, *PDE8B*), cGMP catabolic genes (*PDE5A*, *PDE6A-B*, *PDE6D*), or dual cAMP/cGMP catabolic genes (*PDE1A-C*, *PDE3A-B*, *PDE10A*), may sensitize NSCLC cells to PMX. Earlier work by our group also suggests that *PDE9A* mutations may have a similar effect [16]. Those findings may be particularly relevant to how PMX is used in the maintenance setting for NSCLC, where the antimetabolite is administered as a single agent with conflicting reports on clinical benefits [55,56,57,58]. To our knowledge, our study is the first to describe a strong association between cAMP/cGMP catabolic genes and response to PMX, supporting the use of biochemical PDE inhibitors to overcome PMX resistance in PDE^wt^ tumors.

The clinical potential of the PDEi-PMX combination comes with important caveats. A strong growth-stimulating effect of single-agent PMX was observed in two AD cell lines (H1781 and Calu3) and two SCC cell lines (H1703 and Ludlu1). In three of these four cell lines (SCC cell lines H1703 and Ludlu1; AD cell line H1781), combination treatment with any of the four PDEis demonstrated cytotoxic effects. Those findings are consistent with earlier reports showing that PDE5i can promote the cytotoxicity of PMX [52,59]. Importantly, whereas low-dose PMX (0.025 µM) combined with PDEis acted synergistically to promote H1781 cytostasis and H1703 and Ludlu1 apoptosis, high-dose-PMX combinations largely diminished anti-SCC activity and promoted AD H1781 cell growth through strong antagonistic interactions. It has previously been shown that high-dose PMX promotes cellular senescence, which contributes to a strong synergistic anti-AD activity when combined with gefitinib, a first-generation EGFR tyrosine kinase inhibitor [60]. We show that both H1781 and Calu3, the two AD cell lines most responsive to single-agent PMX in our study, enter a reversible pre-apoptotic state in significant proportion when exposed to high-dose PMX. This physiological state, akin to a cellular autophagy-/-senescent state, may be a contributing factor to the strong antagonistic interaction we observed with H1781 AD cells when combining metabolic inhibitors (PDEis) with an anti-folate agent. Microdosing PMX to achieve maximum synergistic activity with PDE biochemical inhibition, at a dose 10,000 lower than that which is clinically utilized, may present its own challenges considering that sublethal dosing of cytotoxic chemotherapy can promote the acquisition of drug resistance in lung cancer cells in vitro [61,62]. However, the concentration of PDE9i-10i employed in our study may still be clinically achievable. Previous clinical studies have demonstrated that serum concentrations of 48.5–59.5 ng per mL and 17.6–36.5 ng per mL for PDE9i and PDE10i, respectively, are well tolerated clinically, with no reported adverse events (Appendix A). The same cannot be said for PDE5i (sildenafil), for which we observed anti-NSCLC activity at 10,000 ng per mL, far above the recommended clinical dose (50 mg, i.e., 159–531 ng per mL serum titer, Appendix A). Higher sildenafil doses can have serious and potentially life-threatening side effects, including blindness and serious cardiovascular or cerebrovascular events [63,64]. Five deaths have been linked to a sildenafil overdose (870–1300 ng/mL in blood serum) [64]. Certainly, this raises safety concerns for targeting PDE5 with sildenafil in NSCLC.

## 5. Conclusions

PMX’s single-agent activity is associated with cAMP/cGMP catabolism. Genomic alterations to PDE genes or biochemical inhibition of PDE function can sensitize NSCLC to PMX in vitro. Although single-agent PDE inhibitors potently stimulate growth in NSCLC, combining PDE inhibitors (PDE8i, PDE5i, PDE9i, and PDE10i) with low-dose PMX may enhance the anti-NSCLC activity of PMX in both squamous and non-squamous NSCLC cell lines. The combination of PMX-PDEi may have a very narrow therapeutic window against NSCLC, challenging the clinical ability to achieve and maintain that dose level in a clinical setting. Indeed, higher-dose PMX completely reverts the drug combination to a potent growth-stimulating/anti-apoptotic regimen.

## Figures and Tables

**Figure 1 cancers-16-02475-f001:**
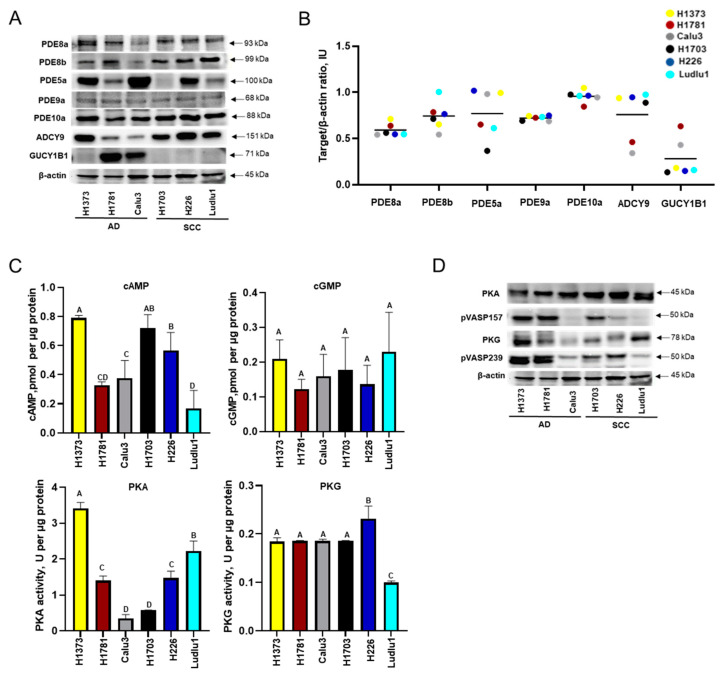
Baseline characteristics of non-small-cell lung cancer (NSCLC) cell lines. Untreated lung cancer adenocarcinoma (AD) cell lines (H1373, H1781, and Calu3) and lung cancer squamous-cell carcinoma (SCC) cell lines (H1703, H226, and Ludlu1) in exponential phase of growth were screened. (**A**) Protein expression of five isoforms of enzymes involved in cyclic nucleotide catabolism, PDE8a, PDE8b, PDE5a, PDE9a, and PDE10a, and two anabolic enzymes, adenylate cyclase (ADCY9) and guanylate cyclase (GUCY1B1), assessed by Western blotting. β-actin was used as the loading control. (**B**) Cyclic nucleotide catabolic and anabolic enzymes in NSCLC cell lines. Quantification of the ratio of cyclic nucleotide metabolic pathway proteins versus β-actin in NSCLC cell lines. Optical density of the signal is reported in intensity units (IUs). Center lines are in black. (**C**) Basal cAMP and cGMP and direct protein kinase (PKA and PKG) activity determined by cyclic nucleotide ELISA kit and with colorimetric activity kit, respectively. Cellular metabolite concentrations expressed as pmol cyclic nucleotides per µg of cellular protein. Enzymes activity presented as U per µg of cellular protein. Vertical bars represent standard error of the mean (SEM). Different letters indicate values that are significantly different from each other (*p* < 0.05); N = 2, 4, 5. (**D**) Enzymes associated with cyclic nucleotide metabolism in NSCLC cell lines. Protein kinase A (PKA), protein kinase G (PKG), and phosphorylated vasodilator-stimulated phosphoprotein, VASP (Ser157 and Ser259), were measured in six NSCLC cell lines by Western blotting. β-actin was used as the loading control. The original Western blot figures can be found in Appendix A.

**Figure 2 cancers-16-02475-f002:**
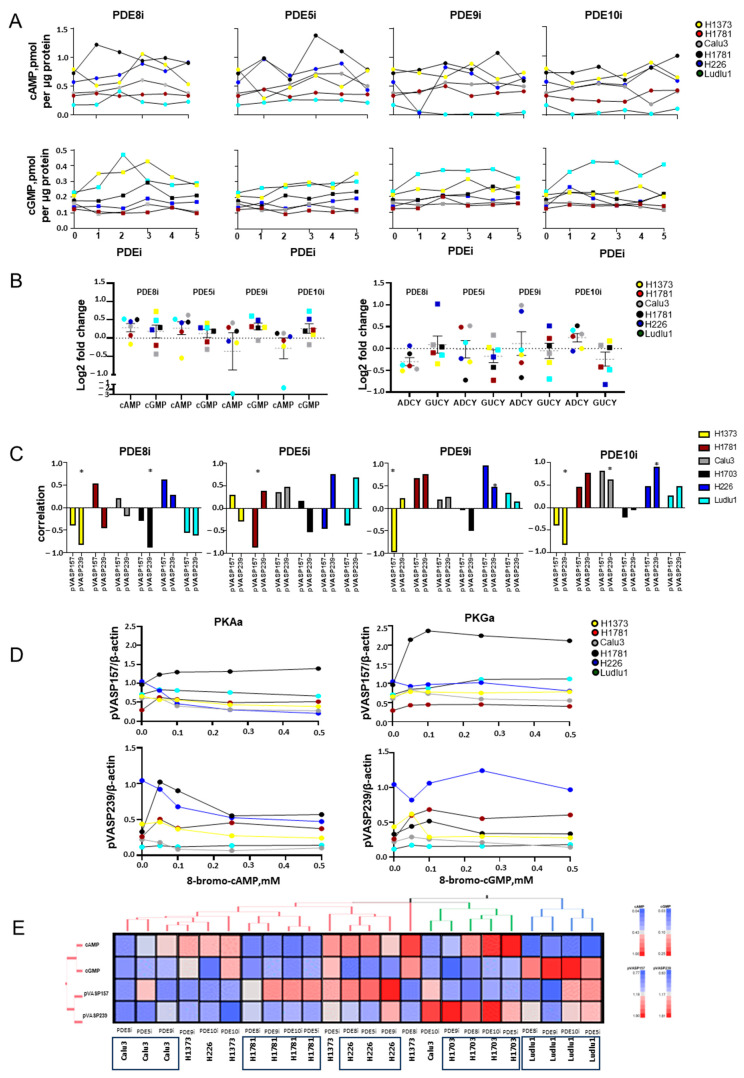
Phosphodiesterase inhibitors modulate cyclic nucleotide signaling in NSCLC cell lines. NSCLC cell lines (H1373, H1781, Calu3, H1703, H226, and Ludlu1) in exponential phase of growth were exposed for 72 h to PDE8i (0–6 nM), PDE5i (0–6000 nM), PDE9i (0–60 nM), or PDE10i (0–6 nM), followed by the following measurements: (**A**) Intracellular cAMP and cGMP levels in NSCLC cells as determined with a cyclic nucleotide ELISA kit. Cellular metabolite concentrations as expressed as pmol cyclic nucleotides per µg of cellular protein. N = 4, 5. (**B**) Simultaneous changes in cAMP and cGMP concentrations; expression level of adenylate cyclase (ADCY) and guanylate cyclase (GUCY) in the presence of PDE inhibitors. Data are presented as log2 fold changes in cAMP or cGMP concentration (left panel) and ADCY or GUCY protein expression level (right panel) between PDEi-treated (all concentrations) versus untreated cells. Center line and upper and lower quartiles for all cells are in grey. (**C**) Effect of PDEis on indirect PKA and PKG activity as determined by measuring pVASP157 and pVASP239 protein levels via Western blotting. β-actin was used as the loading control. Data are presented as correlations between increased concentrations of PDEi and changes in target protein level per cell line. * indicates significant differences (*p* < 0.05). (**D**) Cell-specific activation of PKA and PKG by direct activators (8-bromo-cAMP for PKA or 8-bromo-cGMP for PKG) as determined by phosphorylation of PKA target (pVASP157) and PKG target (pVASP239) measured by Western blotting. Cells were treated with PKA activator (0–1 mM) or PKG activator (0–1 mM) for 72 h. β-actin was used as the loading control. Optical density of the signal is reported in intensity units (IUs). (**E**) Cluster analysis: heatmap profiles of NSCLC exposed to PDEi EC50 with cAMP, cGMP, pVASP157, and pVASP239 (N = 24). Columns of the heatmap represent samples with annotated cluster membership. Rows of the heatmap represent metabolites (cAMP or cGMP) or activity of PKA or PKG (pVASP157 or pVASP239, respectively).

**Figure 3 cancers-16-02475-f003:**
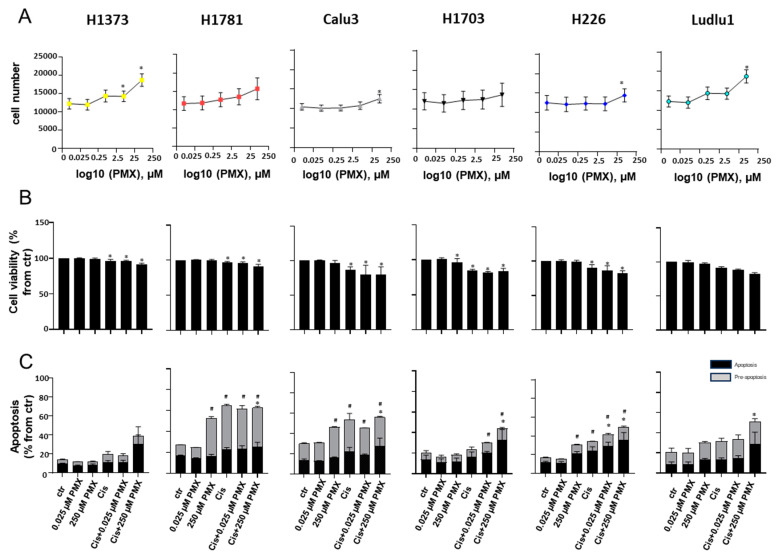
Single-agent pemetrexed shows no anti-tumor activity against PDE^wt^ NSCLC cells. NSCLC cell lines (H1373, H1781, Calu3, H1703, H226 and Ludlu1) in exponential phase of growth were exposed for 72 h to PMX (0.025 µM, 0.25 µM, 2.5 µM, 25 µM, or 250 µM of PMX), cisplatin (2.2 µM), or cisplatin combined with PMX. (**A**) Measurements of single-agent PMX were determined using WST-8 cell-counting kit. Data are presented as cell number, mean ± SEM. N = 34, 37, 39, and 47. (**B**) Measurement of cytotoxic activity of single-agent PMX or cis-PMX combination measured by propidium iodide cell viability assay. Data are presented as % from control, N = 4. (**C**) Measurements of PMX single-agent or cis-PMX combination pro-apoptotic activity are measured using 7-AAD/Annexin-V flow cytometry staining. Data are presented as % total cells acquired, N = 4. * indicates significant differences from control (DMSO-treated); # indicates significant difference for pre-apoptosis from control (DMSO-treated) (*p* < 0.05).

**Figure 4 cancers-16-02475-f004:**
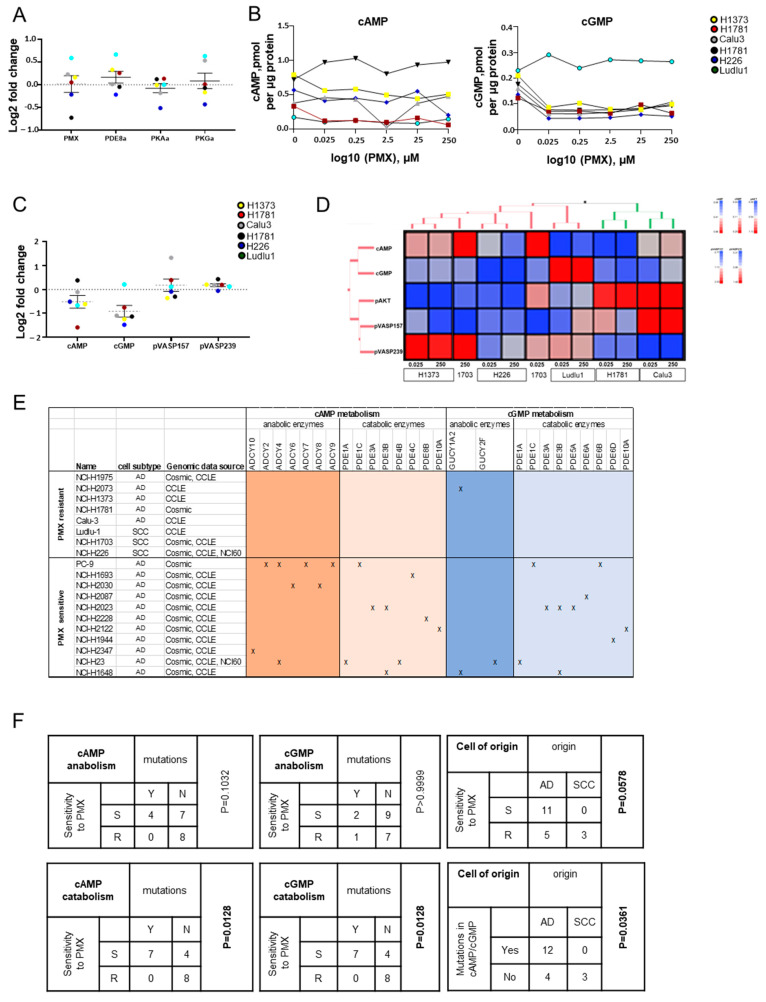
Impairment of cAMP/cGMP catabolism and downstream PKA/PKG signaling are associated with response to PMX in NSCLC. (**A**) NSCLC cell lines (H1373, H1781, Calu3, H1703, H226, and Ludlu1) in exponential phase of growth were exposed for 72 h to PMX (0.025 µM, 0.25 µM, 2.5 µM, 25 µM, or 250 µM of PMX), followed by measurement of pAKT activation. Data are presented as log2 fold changes in phospho-AKT (pAKT (Ser473) level per cell line between all tested concentrations of PMX, PDE8i, PKAa, and PKGa, and non-treated cells were determined by Western blotting. β-actin was used as the loading control. (**B**) Intracellular cAMP and cGMP levels in NSCLC cells after PMX treatment were measured with cyclic nucleotide ELISA kit. Cellular metabolite concentrations are expressed as pmol cyclic nucleotides per µg of cellular protein. N = 4, 5. (**C**). Effect of PMX on cyclic nucleotide metabolism. cAMP and cGMP were measured using cyclic nucleotide ELISA kit. Indirect PKA and PKG activity was determined by measuring site-specific phosphorylation of vasodilator-stimulated phosphoprotein; PKA (pVASP157) and PKG (pVASP239) activity was determined by Western blotting. β-actin was used as the loading control. Data are presented as log2 fold changes in target level per cell line between all tested concentrations of PMX and non-treated cells. (**D**) Cluster analysis: heatmap profiles of NSCLC exposed to PMX with cAMP, cGMP, pVASP157, pVASP239, and pAKT (N = 30). Columns of the heatmap represent the samples with annotated cluster membership. Rows of the heatmap represent the metabolites (cAMP or cGMP), activated AKT (pAKT), or activity of PKA or PKG (pVASP157 or pVASP239, respectively). (**E**) Association between cAMP/cGMP metabolic enzyme mutation status and NSCLC sensitivity to PMX in vitro. Single-agent PMX activity was measured using colony formation (those unpublished data were used with the authorization of Dr. Misty Shields and Dr. John Minna) [33]. Mutation status of genes coding for cAMP/cGMP anabolic or catabolic enzymes was assessed using the Cosmic, CCLE, or NCI60 databases. **x** indicates the presence of non-synonymous mutations. (**F**) Contingency analyses (Fisher’s exact test) comparing NSCLC histology (AD, SCC), cAMP anabolic/catabolic enzyme mutation status, or cGMP anabolic/catabolic enzymes mutation status (N = 19) with PMX sensitivity.

**Figure 5 cancers-16-02475-f005:**
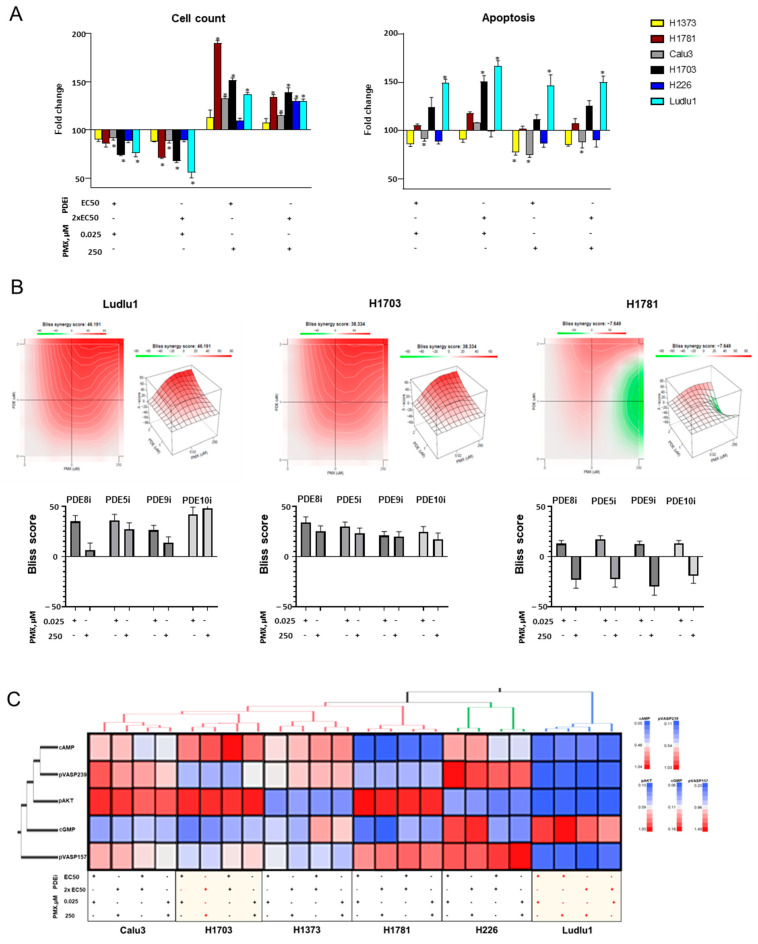
PDEi can sensitize PDE^wt^ NSCLC cells to low-dose PMX killing. NSCLC cell lines (H1373, H1781, Calu3, H1703, H226, and Ludlu1) in exponential phase of growth were exposed for 72 h to PMX (untreated, 0.025 µM or 250 µM) and EC50 or 2xEC50 PDEi, followed by the following measurements: (**A**) Cellular growth and apoptosis: combined pMX and PDEi treatment was associated with cell growth inhibition in some NSCLC cell lines. Caspase 3/7 activity was measured to evaluate PMX and PDEi induced apoptosis. Cell growth and apoptosis compared to the corresponding untreated cells are shown. Data from individual inhibitors were combined for each studied NSCLC cell lines. Data are presented as fold change (%) from control. N = 8. * indicates significant differences from untreated respective control (*p* < 0.05). (**B**) Drug interaction matrices were created for Ludlu1, H1703, and H1781 cells based on Bliss synergy scoring using combined PMX and PDEi treatment (drug synergy shown in red, drug antagonism shown in green). Bliss synergy scores for PDE8i, PDE5i, PDE9i, and PDE9i were individually plotted with low (0.025 µM) and high (250 µM) PMX doses (Bliss score > 10 indicates synergy; Bliss score < −10 indicates antagonism). (**C**) Cluster analysis: heatmap profiles of NSCLC exposed to 0.025µM or 250 µM PMX and PDEi (EC50 or 2xEC50) with cAMP, cGMP, pVASP157, pVASP239, and pAKT (n = 24). Columns of the heatmap represent the samples with annotated cluster membership. Rows of the heatmap represent the metabolites (cAMP or cGMP) activated AKT (pAKT) or activity of PKA or PKG (pVASP157 or pVASP239, respectively).

## Data Availability

The authors declare that the data supporting the findings of this study are available within the article and its Appendix A. Source data are provided with this paper.

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
