# Peer review of "Phosphodiesterase Inhibition to Sensitize Non-Small-Cell Lung Cancer to Pemetrexed: A Double-Edged Strategy"

_cancers, 2024, doi:10.3390/cancers16132475_

Round 1

Reviewer 1 Report

Comments and Suggestions for Authors

The authors suggest that although single-agent PDE inhibitors are potent pro-mitotic agents in NSCLC, combining PDE inhibitors (PDE8i, PDE5i, PDE9i, and PDE10i) with low-dose PMX may enhance the anti-NSCLC activity of PMX in both squamous and non-squamous NSCLC cell lines. However, surprisingly, this result was not obtained with higher doses of PMX.

Major points

1. As shown in Figures 2b and 2c, the effects of Phosphodiesterase inhibitors on cyclic nucleotide signaling vary considerably between cells. This suggests that Phosphodiesterase inhibitors do not have a universal impact on NSCLC.

2. In Figure 3a, the number of cells does not decrease even as the concentration of PMX increases. What could be the reason for this experimental outcome?

3 The most crucial reason for the lack of difference between the results of low and high concentrations of PMX has not been demonstrated.

Comments on the Quality of English Language

The authors suggest that although single-agent PDE inhibitors are potent pro-mitotic agents in NSCLC, combining PDE inhibitors (PDE8i, PDE5i, PDE9i, and PDE10i) with low-dose PMX may enhance the anti-NSCLC activity of PMX in both squamous and non-squamous NSCLC cell lines. However, surprisingly, this result was not obtained with higher doses of PMX.

Major points

1. As shown in Figures 2b and 2c, the effects of Phosphodiesterase inhibitors on cyclic nucleotide signaling vary considerably between cells. This suggests that Phosphodiesterase inhibitors do not have a universal impact on NSCLC.

2. In Figure 3a, the number of cells does not decrease even as the concentration of PMX increases. What could be the reason for this experimental outcome?

3 The most crucial reason for the lack of difference between the results of low and high concentrations of PMX has not been demonstrated.

Author Response

Please see the attachment with the answers for Reviewer #1

Reviewer 2 Report

Comments and Suggestions for Authors

* The authors investigated the effects of Phosphodiesterase inhibitors on the increased sensitivity of non-small cell lung cancer to chemotherapeutic agents like pemetrexed. The investigation was well designed and the manuscript was well-written. PDEi had no cytotoxic or cytostatic effect on six selected NSCLC cell lines. PDEi5, PDE8-10i enhanced the effects of pemetrexed on non-small cell lung cancer. This effect was independent of NSCLC histology or baseline cAMP/PKA, cGMP/PKG signaling. Furthermore, the authors observed that PDEi/PMX synergistic activity was only present in the combination of micro-dosing of pemetrexed, whereas PDEi in combination with a higher dose of PMX reverted to pro-mitosis.

* The introduction was well written. In the material and methods, all catalog numbers should be mentioned for all used kits. The results were clearly displayed and well discussed. 

* Overall, the study is well-designed and well-written, and I don't have any additional comments.  

Comments on the Quality of English Language

* English revision should be done to correct few mistakes such as line 22: PDE5 should be corrected into PDE5i. All materials and methods and results must be written at past. 

Author Response

Please see the attachment with the answers for Reviewer #2

Reviewer 3 Report

Comments and Suggestions for Authors

Here Foureau et al reported that phosphodiesterase inhibition sensitize non-small cell lung cancer cells to pemetrexed treatment without direct cytotoxic or cytostatic anti-NSCLC activity in vitro. On the whole, this article is clearly presented and interesting, but I have some comments.

1. This study is only focused on in vitro experiments showing DE inhibitors/PMX synergistic activity only in combination of micro dosing of PMX. Additional experiments are needed to confirm the efficacy of combination of PDE inhibitors and PMX in animal models.

2. Do PDE inhibitors have suppressive influence on the growth of normal human bronchial epithelial cells such as BEAS-2B?

3. It is interesting to know whether the growth and apoptosis of human cells with molecular features, including KRAS or EGFR mutation. After all, PMX-based chemotherapy is recommended in the linical management of NSCLC with or without KRAS/EGFR mutations.

4. Why high-dose PMX (250 μM) was employed, the cytotoxic/cytostatic effect of combination PDEi with PMX treatment was abrogated and even presented an anti-apoptotic effect in NSCLC cells? Additional explanation and discussion of molecular mechanism and potential clinical application should supplemented.

Comments on the Quality of English Language

Minor editing of English language required.

Author Response

Please see the attachment with the answers for Reviewer #3

Reviewer 4 Report

Comments and Suggestions for Authors

The study evaluated the biochemical inhibition of PDE5,  alone or in combination with PMX in squamous and non-squamous NCSLC cell lines. PMX/PDEi5, PDE8-10i combinations promoted apoptosis in half NSCLC cell lines evaluated and the synergistic PMX/PDEi anti-NSCLC activity was independent of NSCLC histology but require micro dosing of PMX.

The article has strong data, well well-designed. I would just improve the abstract which is not well structured and defined about background, methodology and results. I would also give more information about the genetic background of the cell lines. And to finish clarify the limitations of the study. 

Revise text for minor spelling mistakes and grammar issues. 

Comments on the Quality of English Language

minor mistakes

Author Response

Please see the attachment with the answers for Reviewer #4
